# Hierarchical LSTM-Based Network Intrusion Detection System Using Hybrid Classification

**Jonghoo Han and Wooguil Pak \***

Department of Information and Communication Engineering, Yeungnam University,
Gyeongsan 38541, Republic of Korea
\* Correspondence: wooguilpak@yu.ac.kr; Tel.: +82-53-810-3092

**Abstract:** Most existing network intrusion detection systems (NIDSs) perform intrusion detection using only a partial packet data of fixed size, but they suffer to increase the detection rate. In this study, in order to find the cause of a limited detection rate, accurate intrusion detection performance was analyzed by adjusting the amount of information used as features according to the size of the packet and length of the session. The results indicate that the total packet data and all packets in the session should be used for the maximum detection rate. However, existing NIDS cannot be extended to use all packet data of each session because the model could be too large owing to the excessive number of features, hampering realistic training and classification speeds. Therefore, in this paper, we present a novel approach for the classifier of NIDSs. The proposed NIDS can effectively handle the entire packet information using the hierarchical long short-term memory and achieves higher detection accuracy than existing methods. Performance evaluation confirms that detection performance can be greatly improved compared to existing NIDSs that use only partial packet information. The proposed NIDS achieves a detection rate of 95.16% and 99.70% when the existing NIDS show the highest detection rate of 93.49% and 98.31% based on the F1-score using two datasets. The proposed method can improve the limitations of existing NIDS and safeguard the network from malicious users by utilizing information on the entire packet.

**Keywords:** hybrid classifier; network intrusion detection; hierarchical LSTM; dual LSTM

## 1. Introduction

As the size and amount of traffic in the network increase, the scale of damage caused by network intrusion also increases. Network intrusions continue to increase despite the development of various technologies to detect and prevent network intrusions and the proposal of many security devices. This is primarily because the number of vulnerabilities used by network intrusions and the polymorphism of the intrusions themselves have increased, making network intrusion detection very difficult. In particular, recent network intrusions actively utilize zero-day attacks, and network intrusion detection technologies designed based on the knowledge of existing attacks often fail to prevent new attacks.

One of the most effective ways to solve this problem involves utilizing machine learning to detect intrusions using the behavior of the intrusion rather than its pattern [1]. Early network intrusion detection systems (NIDSs) using machine learning used conventional machine learning algorithms such as decision trees and random forests, but recent NIDSs are actively adopting deep learning. Early NIDSs classified traffic into sessions, analyzed each session to derive various statistical characteristics, and used them to learn machine learning models and detect network intrusions [2–6]. However, recent deep learning-based NIDSs directly derive features using deep learning from the traffic itself without preprocessing the traffic [7].

This approach has an advantage over the method using existing session features in that features need not be designed and it is not necessary to wait for a session to end.

However, when packet data are used as features for deep learning, the number of features increases according to the amount of packet data; therefore, modern packet data-based NIDS use a fixed size of the front part of the packet data rather than using the entire packet data. According to previous studies, it has very high detection accuracy, even when using a part of packet data [7]. Nevertheless, it is unclear as to whether it is appropriate to use partial packet data.

The use of partial packet data is not a method for optimizing detection performance as described above, but it is advantageous to design a machine learning model when the size of the input dimension of the machine learning model is fixed. Therefore, we must answer the following questions:

(1) Can the detection performance be maximized by intrusion detection using only a part of packet data?
(2) If a part of the packet data may be used, what size is optimal?

This study presents the answers to these questions through various experiments. In addition, these answers are used to propose a model that responds to various packet sizes and lengths by resolving the limitations of the existing packet data-based method. Thus, we propose a new method for designing a deep learning model that can significantly improve the performance of intrusion detection methods using existing packet data.

The proposed method uses a hierarchical model comprising two long short-term memory (LSTM) models [8]. The first LSTM model consists of a dual LSTM model and generates features using the entire packet data [9]. The dual LSTM is designed to efficiently process the entire packet data by accepting various packet lengths as input and minimizing zero-padding that degrades the classification performance. This approach facilitates the creation of high-quality packet features by minimizing the loss of packet data.

The second LSTM processes sessions instead of single packets. The features of the packets included in the same session are sequentially input to each cell of the LSTM. Similar to the dual LSTM for packets, the second LSTM can handle variable session lengths rather than a fixed number of packets. Unlike conventional methods, the session LSTM avoids performance degradation owing to zero-padding and determines whether a session is malicious by using packet features derived from the packet LSTM. In short, NIDS using existing packet data has many vulnerabilities by using only a small fraction of the data of session traffic, but the proposed NIDS has the ability to use all packet data in the session, enabling sophisticated detection for various classes. This is the innovative strength of the proposed NIDS.

Therefore, the contributions of the proposed method are as follows.

1. The lengths of the packet and session required to maximize the detection performance of the NIDS that directly uses packet data is presented;
2. A method to increase detection accuracy by using the entire packet data, rather than a part, for intrusion detection is proposed;
3. An NIDS model that maximizes performance for per-session intrusion detection by processing sessions of various lengths, rather than sessions of fixed length, is presented.

The remainder of this paper is organized as follows. In Section 2, existing research is reviewed, and in Section 3, the proposed method is presented. Section 4 compares and analyzes the performance of the proposed and existing methods. Finally, Section 5 concludes the paper.

## 2. Existing Work

Most modern NIDSs that apply machine learning classify traffic into logical groups called sessions, calculate the statistical characteristics for sessions, and use them as features to learn machine learning models [10–15]. Therefore, the features created in this manner are referred to as session features. The storage burden is very high because most session features are created after collecting from the first to the last packet of a session.

This problem can become further complicated depending on how each field of the session feature is defined. Each dataset defines session features in a different manner. For example, the features used in the dataset created by UNB consist only of those obtained by analyzing one session [12–15]. Therefore, when a session feature is created in this manner, the NIDS can store each session, analyze the saved session independently, create a feature, and then discard the stored session data. Therefore, the session stored in the NIDS is the only currently active session.

However, in the case of the KDD'99 or Kyoto2006 datasets, the currently active session, as well as the sessions related to them (even terminated sessions existing within a specific time window and with the same destination or the same service in sessions), are also required to create a feature [11,16–20]. Therefore, in this case, compared to the UNB dataset, more session information needs to be stored and related information must be managed; hence, more storage space is required.

Session features basically reflect the characteristics of the entire session; therefore, if the characteristics of intrusion exist in a part of the session rather than the entire session, they may not be clearly revealed and may be diluted by other parts of the session. In particular, there is a fatal disadvantage in that an attacker can easily change the entire characteristics of a session, such as by generating repeated packets in the session and thus, easily evade the NIDS based on session features.

Previous studies have also proven that an attacker can bypass NIDS detection by intentionally adjusting the total traffic volume and length of an attack session. For example, Two-Stage Classifier Ensemble for Intelligent Anomaly-Based Intrusion Detection System (TSE), one of the representative session-based NIDS, is a method that increases the accuracy of machine learning through an algorithm that elaborately selects session features [21]. However, many studies on the vulnerabilities of this approach to NIDS are being conducted [22,23].

Unlike session features, packet data can be used as features (called packet features) [7]. This approach uses multiple packets within a session to create features, rather than a single packet. Therefore, the intrusion detection performance and memory space required to create a feature vary depending on the number of packets used to create the feature. Because the number of packets should not be too small to improve the detection performance, a packet feature-based NIDS requires a large amount of storage space. In particular, since the size of one packet is increasing, even if a feature is created using only one packet, sometimes it cannot be used as a whole. Therefore, in previous studies, features were created using only a part of the packet instead of the entire packet [7].

If the session feature uses a feature that includes the overall characteristics of the session, the packet feature cannot use the entire traffic as a feature because of its characteristics, which makes it impossible to reflect the overall characteristics of the session. Consequently, the packet feature cannot contain all the characteristics of the session compared to the session feature. Therefore, the risk that an attacker can circumvent intrusion detection by embedding malicious code in a part excluded from feature creation remains.

Currently, research on NIDS using packet features is not actively progressing compared to NIDS using session features. HAST-IDS, a representative packet feature-based NIDS, uses a deep learning model implemented based on LSTM and CNN. Packet data uses data for one session, but each packet uses 100 bytes, and since only the first 6 packets of a session are used, most of the traffic is not used for intrusion detection in case of a long session.

Table 1 compares the pros and cons of the session and packet features. Ultimately, to improve the accuracy of an NIDS, an approach that reduces the burden on storage space while including the characteristics of the entire session is essential.

**Table 1.** Comparison of advantages and disadvantages of session and packet features.

| Feature Type | Pros. | Cons. |
|---|---|---|
| Session feature | It reflects the entire properties for each session well. | Since all traffic for each session is required to create a session, a lot of storage space is required. |
| Packet feature | There is no need for an analysis process to create features. | Because of the heavy burden on storage space, the entire traffic cannot be used as a feature. Vulnerable to bypass attacks by using some session traffic. |

### 3. Proposed Algorithm

In this section, the motivation for developing the proposed NIDS and the results of preliminary experiments are first introduced, and the deep learning model used in the proposed NIDS is examined in detail. For better understanding, let us present all abbreviations and definitions in Table 2 before further explanation.

**Table 2.** Each abbreviation in the paper and its definition.

| Abbreviation | Definition |
|---|---|
| NIDS | Network Intrusion Detection System |
| LSTM | Long Short-Term Memory |
| TSE | Two-Stage Classifier Ensemble for Intelligent Anomaly-Based Intrusion Detection System |
| DNN | Deep Neural Network |
| CNN | Convolution Neural Network |
| RNN | Recurrent Neural Network |
| ELM | Extreme Learning Machine |
| ROC | Receiver Operating Characteristic |

*3.1. Motivation*

Unlike existing deep neural networks (DNNs) and convolutional neural networks (CNNs), an LSTM is advantageous in generating outputs of the same shape, even when the input shapes are of various sizes [24,25]. This is because an LSTM extends the structure of the existing recurrent neural network (RNN) that generates outputs of the same dimension for inputs of various sizes, as shown in Figure 1. This characteristic of LSTM allows NIDS to divide each packet by a fixed size and then enter each part into each cell of the LSTM, so that all packets of various sizes can be used as input to the classifier.

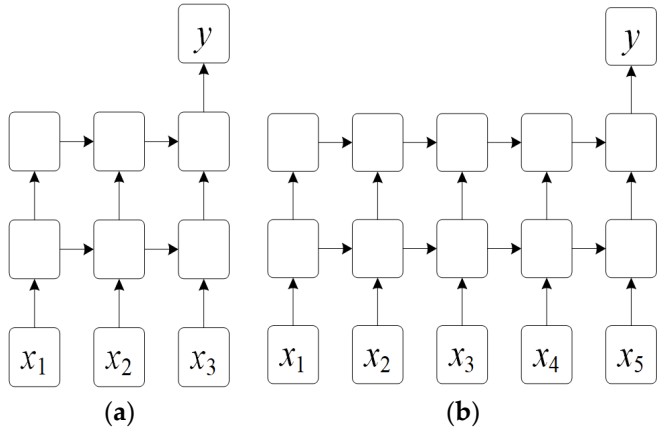

**Figure 1.** Dual LSTM structure is flexible in terms of the number of inputs unlike DNN in which the neural network topology is fixed according to the number of existing inputs. (**a**) LSTM structure when three inputs are given; (**b**) LSTM structure when five inputs are given.

Figure 1a shows a dual LSTM structure in which three identical-sized data are entered into each cell of the LSTM first layer, the output of each cell is entered into each cell of the LSTM second layer, and finally the output of the last cell is used as a classification result for the entire input data. This classifier can freely adjust the number of inputs, so even if the number of data inputs increases to 5, as shown in Figure 1b, the classification can be performed while maintaining the same structure without any modification.

An LSTM can be used for classification using the entire packet data, but no studies have analyzed the relation between packet data size and classification performance. Therefore, a verification experiment is absolutely necessary to confirm that the detection accuracy is improved by using the entire packet data as an input. In this research, the classification performance according to the length of the packet used in NIDS is analyzed. In addition, the classification performance according to the number of packets used in NIDS among the packets belonging to the session is also investigated.

Figures 2 and 3 show the measurement results of the classification performance according to the number of cells used when packet data are divided into 100-byte units and used as input data for each cell. The figures indicate that the overall classification performance increases with the number of cells. This result means that the performance degradation due to the gradient vanishing problem occurs as the number of LSTM cells increases, but the performance increases due to the increase in the data used for classification overwhelms the performance degradation [26]. Therefore, all packet data must be used, unlike previous studies that used only partial packets.

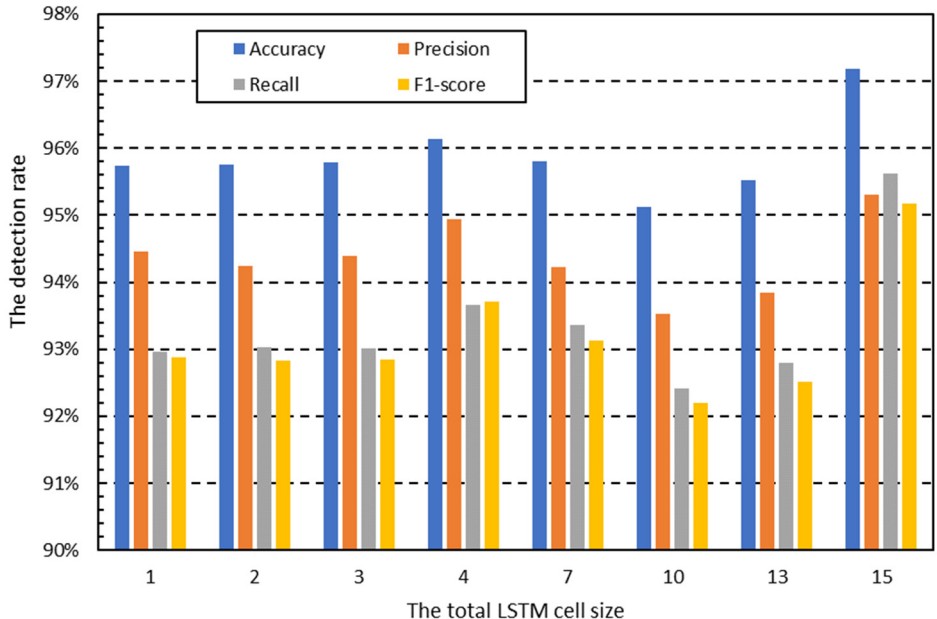

**Figure 2.** Detection rate of the LSTM according to the total LSTM cell size for classifying a packet with ISCXIDS2012.

Figures 4 and 5 show the results of evaluating classification accuracy when all the traffic of each packet is used as the input for the LSTM using the output value used for classification. These results indicate that classification performance increases as more packets of a session are used for classification. Therefore, according to this experiment, classification performance may be improved by using as many packets as possible for the classification of a single packet. Furthermore, as many packets as possible should be used for a single session.

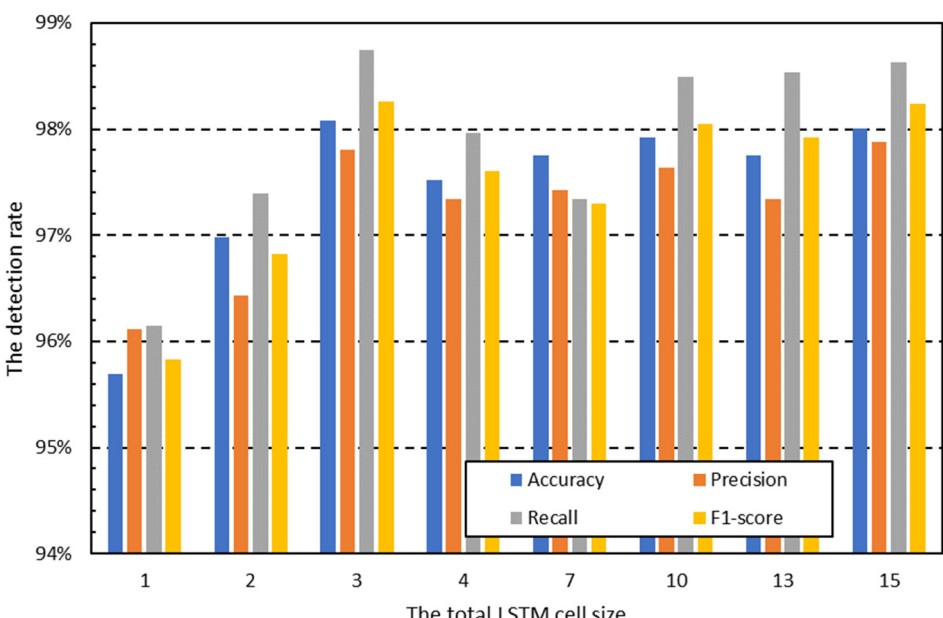

**Figure 3.** Detection rate of LSTM according to the total LSTM cell size for classifying a packet with the CIC-IDS2017 dataset.

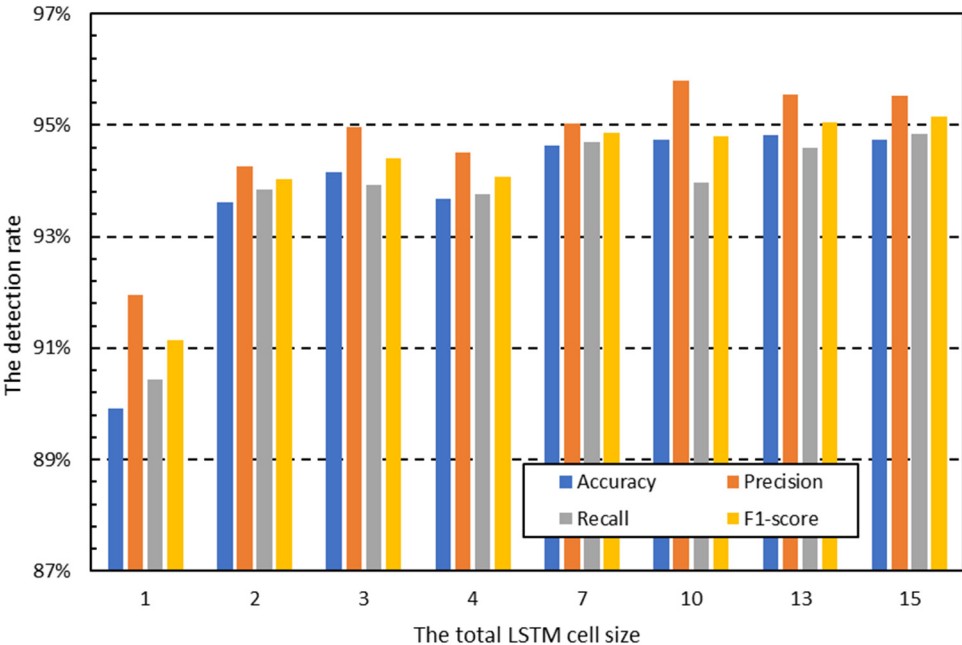

**Figure 4.** Detection rate of LSTM in terms of session according to the total LSTM cell size with the ISCXIDS2012 dataset.

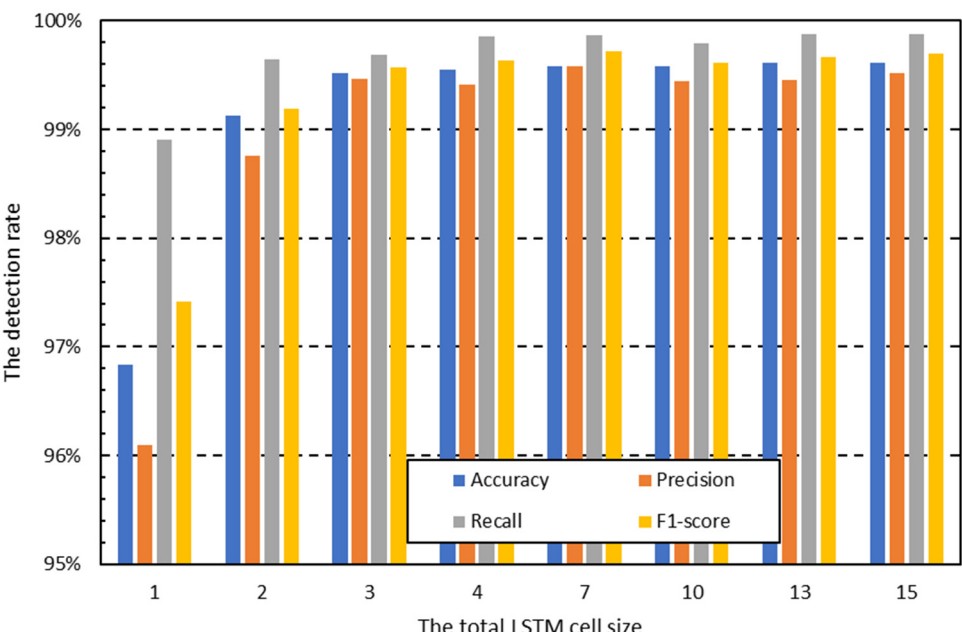

**Figure 5.** Detection rate of LSTM in terms of session according to the total LSTM cell size with the CIC-IDS2017 dataset.

These preliminary experimental results imply that the current design of packet-based NIDS needs to be improved. Existing NIDS uses only some of the initial packets of the session and only partial data for each packet. However, Figure 4 shows that to maximize detection performance, many packets must be used rather than some packets in the session. Figure 5 also shows that the detection performance can be increased only by using the entire packet data rather than some of the packet data. A new approach is therefore needed to address these issues. Considering the results and discussions so far, the following conclusions can be drawn:

- In order to increase the accuracy of network intrusion detection, session traffic should be used as much as possible;
- To this end, the NIDS must use all packet data for each packet and all packets in the session;
- To classify using all data of packets of varying lengths, the packet classifier needs to adopt an LSTM structure;
- To use all packets in a session of various lengths for classification, the session classifier should also be implemented to have an LSTM structure.

### 3.2. Proposed NIDS

According to previous experimental results, the proposed algorithm can be designed as follows: First, the entire packet data, rather than partial packet data, of a received packet should be used as the input to the classifier. Additionally, as many packets as possible in a session should be used as the input to the classifier.

To this end, LSTM is used. However, since LSTM requires the same shape as the input of each cell, one packet must be divided into equal sizes and input to LSTM (i.e., packet classifier). Then, each received packet is converted into a feature for each packet using a packet classifier.

The packet classifier processes packets in a session as a classifier composed of a double LSTM, regardless of the number of packets, and therefore converts each received packet into a packet feature that has the same size regardless of the size of the packet. The packet features converted in this manner are sequentially provided as inputs to the session classifier. Similar to the packet classifier, session classification also consisted of one

LSTM classifier. In this way, regardless of the number of packets in the session, the session classifier can classify the session using the packet features generated from the packets.

Figure 6 shows the entire classification process for the proposed classification model. According to the figure, packet features are created by the packet classifier, and these packet features are integrated into one session feature by the session classifier, and then the final classifier uses it to detect network intrusion. The packet and session classifiers will now be described in detail.

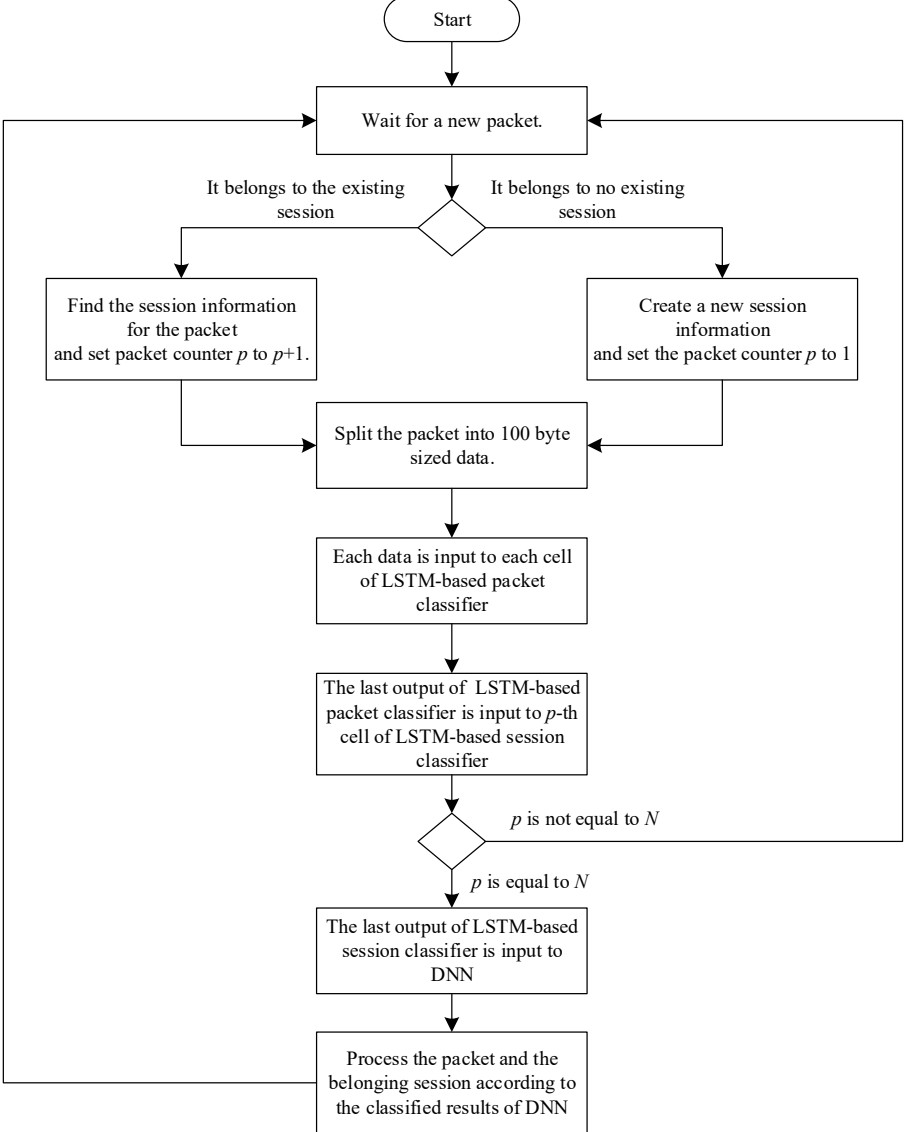

**Figure 6.** The overall classification process for the proposed classification model, where N denotes the total number of cells in LSTM-based session classifier.

### 3.2.1. Dual LSTM-Based Packet Classifier

First, each packet received in the session is divided into 100-byte units, and a portion smaller than 100 bytes is zero-padded and input to each cell of the packet classifier, as shown in Figure 7. The output of the last cell for the packet classifier is used as the input feature of the session classifier. The packet classifier is configured in duplicate to create better quality packet features.

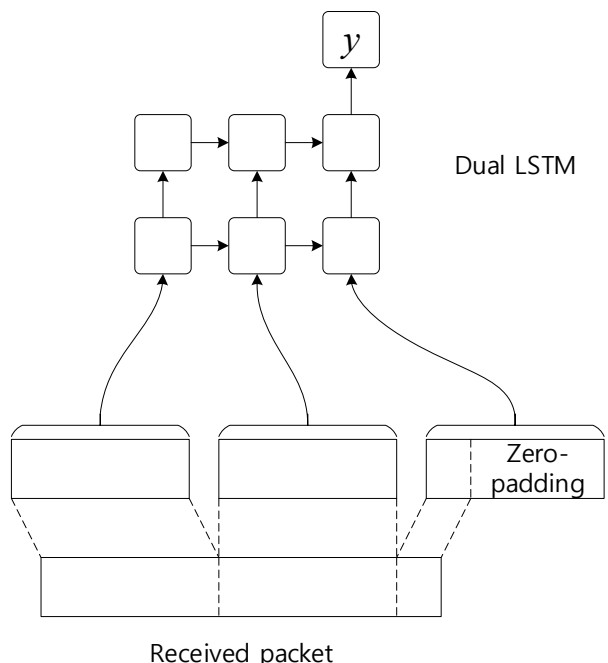

**Figure 7.** Dual LSTM-based packet classifier when the receive packet is partitioned into three inputs to Dual LSTM.

### 3.2.2. LSTM-Based Session Classifier

The session classifier bundles the packet features generated by the packet classifier into one session feature and processes them. Therefore, the input of the $n$-th cell of the session classifier is the output value of the packet classifier for the $n$-th packet of the session. Because the session classifier uses the output value of each cell, it determines whether the corresponding session is intruded upon by using various output combinations. For example, classification may be performed using only the output of the last cell; however, the outputs of all cells may be considered together.

Therefore, to determine the most effective method, the classification performance of various methods was measured based on the CIC-IDS2017 dataset, and the method that achieved the highest performance was selected. To this end, we experimented with different cases: using the output values of all cells of the session classifier, using only the last output value, using the average score for each class of all output values, and selecting the most selected class for all output values. The results of each experiment are listed in Table 3.

**Table 3.** F1 score for each combination method of cell output for the session classifier.

| Combination Type | Train Dataset | Test Dataset |
| :---: | :---: | :---: |
| All cells | 98.7396 | 98.3010 |
| Last cell | 98.7421 | 98.3109 |
| Average score | 98.7396 | 98.3089 |
| Most selected | 98.7396 | 98.3089 |

The results of this experiment confirmed that the highest F1-score was achieved when only the output of the last cell was used in both the training and test datasets. Therefore, according to the experimental results, the proposed method was designed to use only the output value of the last cell of the session classifier. The final output value was provided as an input to the DNN, and the output of the final DNN determined whether to invade. This method has several advantages. First, the overall structure of the classifier is simplified and the classification speed is increased. In addition, when packets being processed belong

to one session, memory space can be saved because it is not necessary to store the output result of the cell for the mid-session packet of the session classifier.

### 3.2.3. Hybrid Classifier Combining Packet and Session Classifiers

Figure 8 shows the overall classification structure of the proposed NIDS. The cell output of each packet classifier is directly input to the corresponding cell of the session classifier without waiting for the next packet, and the output value of the hidden layer for that cell is stored such that the session LSTM continues to classify when the next packet is received. Therefore, the storage space required for each session is only $512 \times 32$-bit floats, or 2048 Bytes. Therefore, unlike the conventional session-based method, which requires all packets to be stored, the proposed method significantly reduces memory usage. In particular, the fixed-size memory used for each session by the proposed method is a significant advantage.

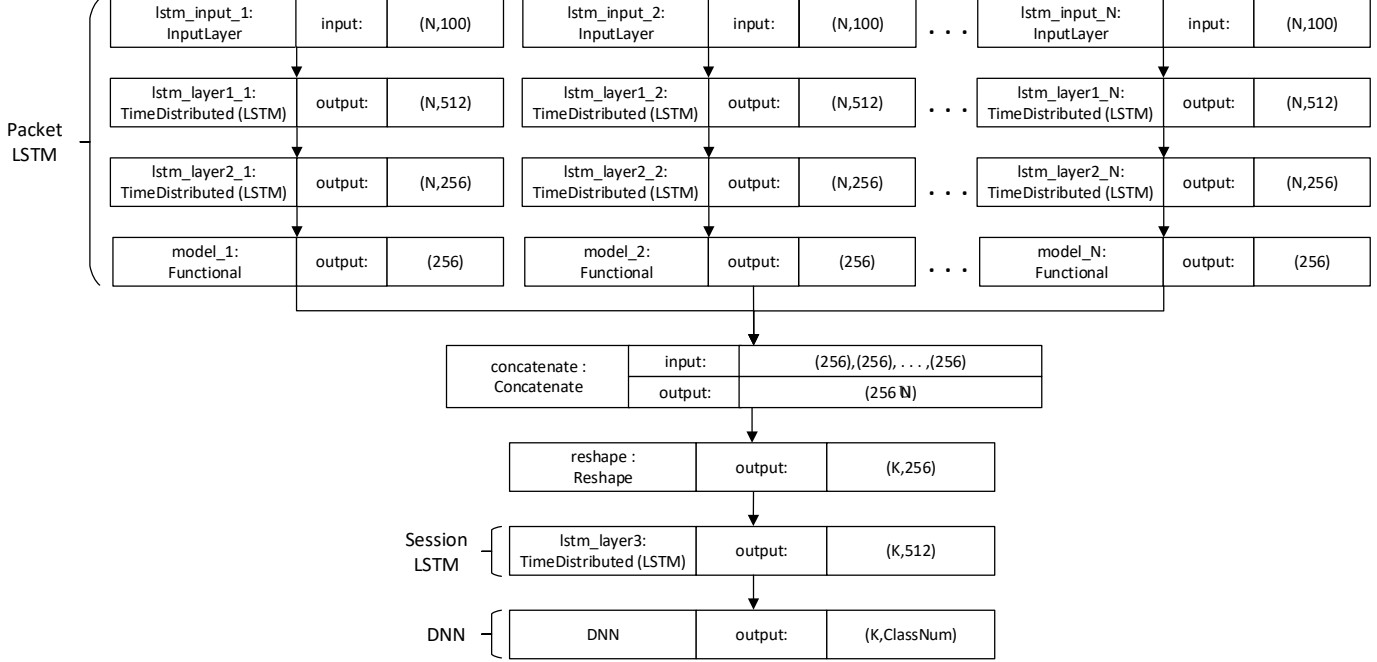

**Figure 8.** Proposed hybrid classifier for NIDS.

The computational complexity $T_{session}$ for classifying one session by the proposed NIDS is given as follows:

$$T_{session} = \sum_{p=1}^{N} L_p / 100 = \Theta\left(\frac{N\overline{L}}{100}\right) = \Theta\left(N\overline{L}\right),$$

where $L_p$ and $\overline{L}$ denote the size of the $p$-th packet in the session and the average packet size, respectively. Therefore, the computational complexity is proportional to the total size of the session traffic, so the computational complexity is higher than that of other competitive NIDSs that require only partial traffic. However, since it has linear complexity, the computational complexity can be reduced by a parallel processing or by increasing the size of the packet partition.

### 3.2.4. Total System Structure

Figure 9 shows the whole NIDS structure equipped with the proposed classifier. As shown in the figure, the hybrid classifier determines whether the session is normal or intrusive, and the result is stored in the session information database. When a new packet is received, this database is used to determine whether or not the packet belongs to a previously classified session. Session information stores the class of the session to which the received packet belongs. Therefore, when information about the session to which the packet belongs is found in session information, the packet is processed according to the stored class. On the other hand, if session information does not exist or if the session has not yet been classified, the packet is delivered to the hybrid classifier. Session information database helps to process received packet belonging to the classified session very fast by discarding it if the session belongs to intrusion class or by forwarding otherwise. Hence, the session information database minimizes the burden of the hybrid classifier and greatly increases the packet processing speed.

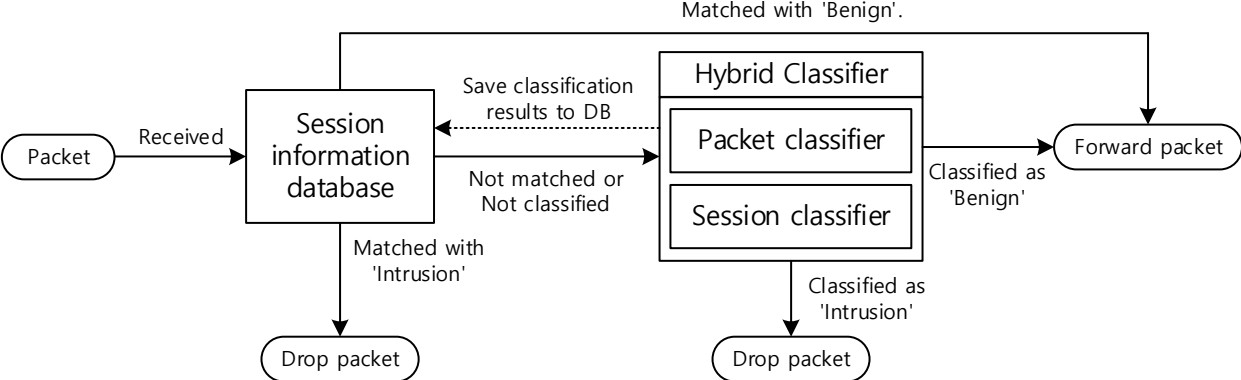

**Figure 9.** Total system architecture for the proposed NIDS.

## 4. Performance Analysis

The ISCXIDS2017 and CIC-IDS2017 datasets were used to accurately analyze the performance of the proposed method. We compared and analyzed the performance with well-known deep learning models, i.e., extreme learning machine (ELM), DNN, CNN, and TSE. Accuracy, precision, recall, and F1-score were the metrics used for analyzing the intrusion detection rate, and each metric is defined as follows:

$$\text{Accuracy} = \frac{TP + TN}{Total},$$

$$\text{Precision} = \frac{TP}{TP + FP},$$

$$\text{Recall} = \frac{TP}{TP + FN},$$

$$\text{F1} - \text{score} = 2\frac{\text{Precision} \cdot \text{Recall}}{\text{Precision} + \text{Recall}},$$

where TP, TN, FP, and FN denote true positive, true negative, false positive, and false negative, respectively.

### 4.1. Dataset Preprocessing

To create packet features to be used for performance evaluation, data for each packet were extracted from the packet capture data included in the ISCXIDS2012 and CIC-IDS2017 datasets. To prevent the machine learning model from being biased towards a specific session, the source IP, destination IP, source port, and destination port fields in the packet were removed. For the session features, the flow ID, source IP, destination IP, and timestamp columns were removed from the ISCXIDS2012 and CIC-IDS2017 datasets for the same reason. Detailed information for each dataset is presented in Table 4.

**Table 4.** Details of the datasets used for performance evaluation.

| Dataset | ISCXID2012 | | CICIDS2017 | |
|---|---|---|---|---|
| Number of features | 88 | | 88 | |
| Number of classes | 5 | | 11 | |
| Number of samples | 78,878 | | 123,236 | |
| Number of each class | Normal | 22,382 | Benign | 27,234 |
| | DDoS | 21,702 | DDoS | 24,860 |
| | BruteForceSSH | 18,145 | DoS Hulk | 20,419 |
| | HTTPDoS | 9487 | DoS GoldenEye | 15,305 |
| | Infiltration | 7162 | PortScan | 6459 |
| | | | FTP-Patator | 6267 |
| | | | SSH-Patator | 6029 |
| | | | DoS slowloris | 5256 |
| | | | DoS Slowhttptest | 4630 |
| | | | Bot | 3546 |
| | | | Web Attack Brute Force | 3231 |

### 4.2. Hyperparameter Configuration

In the proposed method, hyperparameters such as dropout and learning rates were set based on the F1-score. There are many hyperparameter tuning techniques that can be applied to find optimal values for hyperparameters. However, since only two parameters are considered in the proposed NIDS, grid search is enough to find the optimal values of hyperparameters. As shown in Figures 10 and 11, the grid search results show an upward convex shape, so it is easy to find the optimal values. As for the optimal value set, as shown in the figures, the dropout was set to 0.0032 and 0.001 for ISCXIDS2012 and CIC-IDS2017, respectively. The learning rate was set to 0.01 for both datasets.

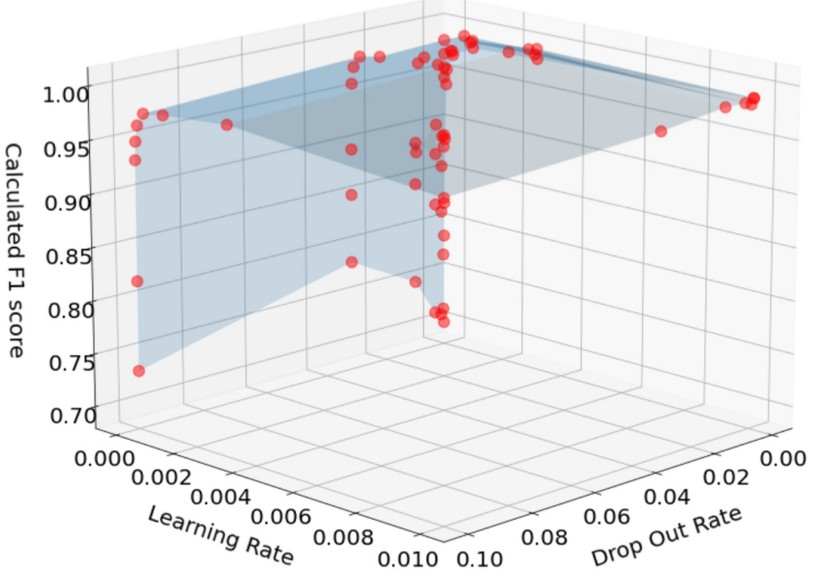

**Figure 10.** Hyperparameter tuning using the ISCXIDS2012 dataset.

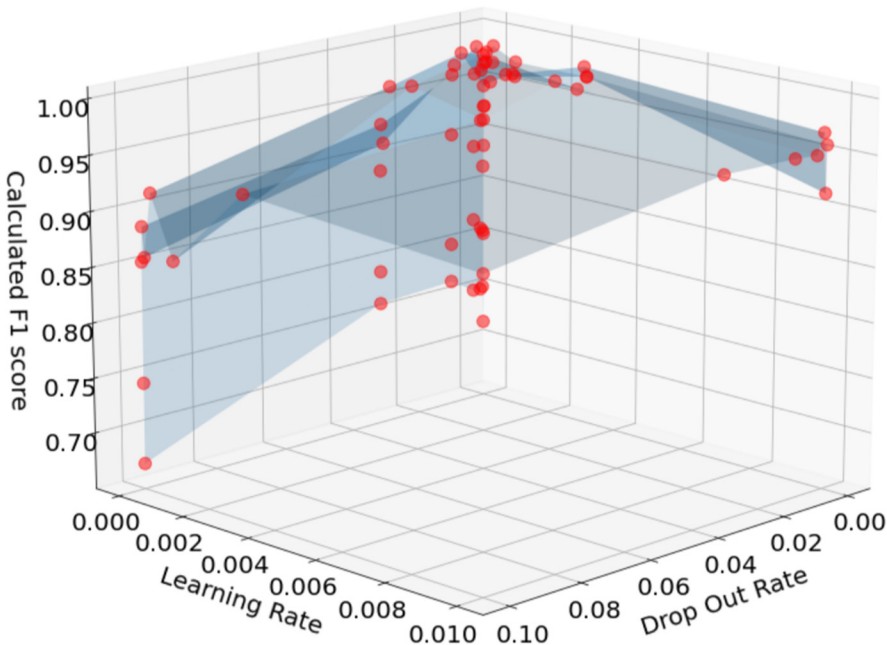

**Figure 11.** Hyperparameter tuning using the CIC-IDS2017 dataset.

### 4.3. Network Intrusion Detection Rate

The detection performances of the proposed algorithm, ELM, DNN, CNN, and TSE are shown in Figures 12 and 13 [27]. The comparison results using the ISCXIDS2012 dataset from Figure 12 confirm the superiority of the proposed method over ELM, DNN, and CNN in terms of all the performance metrics. In particular, the proposed method has a stable performance for all metrics with a very small variation, whereas the performances of the other methods vary for each metric. Thus, false positives and false negatives are few, which confirms that the performance is far superior to those of the existing methods. Among the algorithms used for comparison, DNN and ELM had the best and worst performances, respectively. However, the performance of DNN was 1.67% lower than the proposed method based on the F1-score.

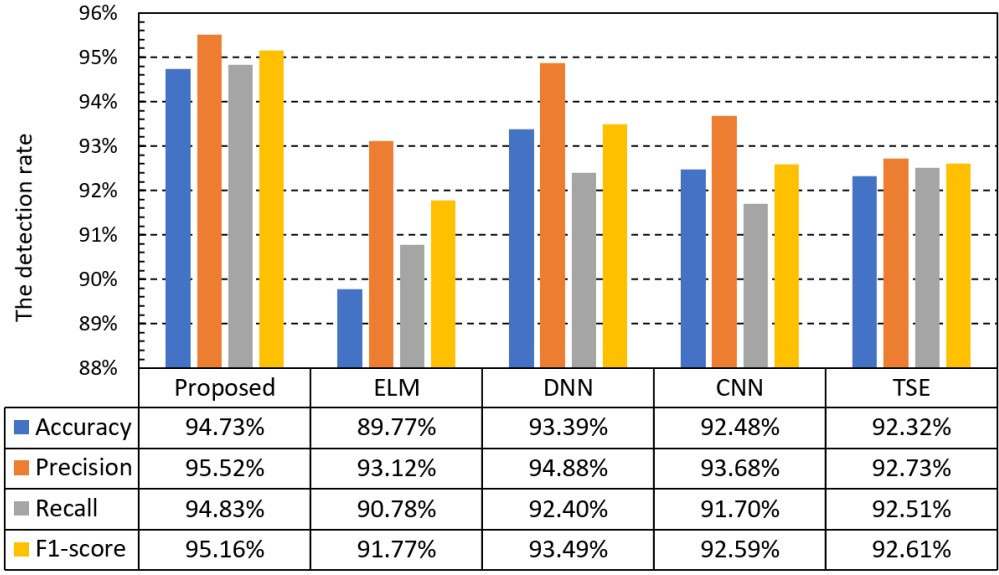

|  | Proposed | ELM | DNN | CNN | TSE |
|---|---|---|---|---|---|
| ■ Accuracy | 94.73% | 89.77% | 93.39% | 92.48% | 92.32% |
| ■ Precision | 95.52% | 93.12% | 94.88% | 93.68% | 92.73% |
| ■ Recall | 94.83% | 90.78% | 92.40% | 91.70% | 92.51% |
| ■ F1-score | 95.16% | 91.77% | 93.49% | 92.59% | 92.61% |

**Figure 12.** Results for the comparison of the intrusion detection rate for the ISCXIDS2012 dataset.

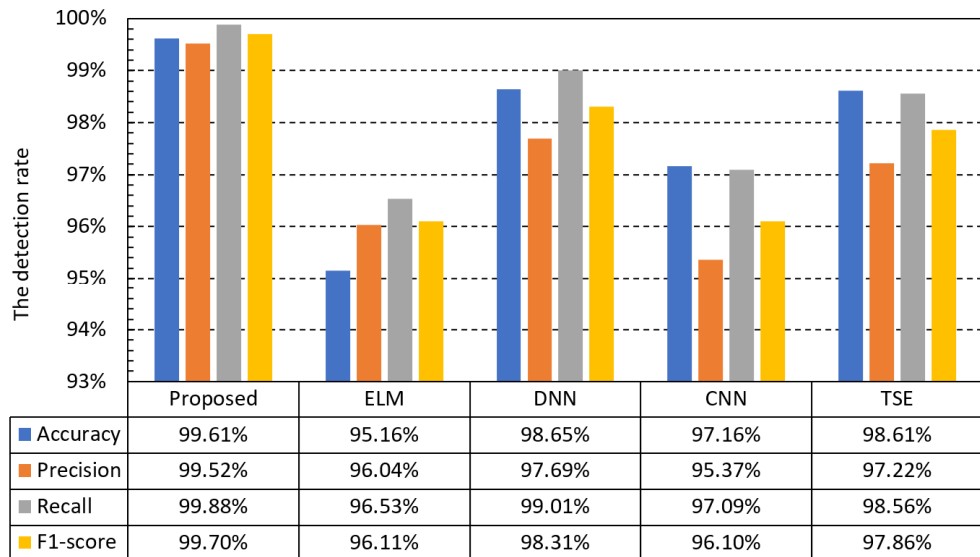

| | Proposed | ELM | DNN | CNN | TSE |
|---|---|---|---|---|---|
| ■ Accuracy | 99.61% | 95.16% | 98.65% | 97.16% | 98.61% |
| ■ Precision | 99.52% | 96.04% | 97.69% | 95.37% | 97.22% |
| ■ Recall | 99.88% | 96.53% | 99.01% | 97.09% | 98.56% |
| ■ F1-score | 99.70% | 96.11% | 98.31% | 96.10% | 97.86% |

**Figure 13.** Results for the comparison of the intrusion detection rate for the CIC-IDS2017 dataset.

The experimental results for the CIC-IDS2017 dataset, shown in Figure 13, are also very similar to those for the ISCXIDS2012 dataset. Although the performance of DNN was the best among comparison methods, it was 1.39% lower than the proposed method in terms of the F1-score. The proposed method exhibited consistently high performance for all metrics. Therefore, regardless of the type of dataset, it exhibited a higher detection rate than other methods while keeping false positives and false negatives low.

Such high detection performance is achieved because the proposed NIDS uses the entire packet data rather than partial data packets for intrusion detection and uses most packets of a session as session features. The creation of fixed-size features that accurately reflect packet information using a double LSTM structure for a large amount of packet data is also considered to have played an important role in achieving stable and high detection performance for all metrics.

### 4.4. Confusion Matrix

The confusion matrices for each dataset are shown in Figures 14 and 15. In the figures, the darker color means the higher value. From Figure 14, it can be observed that the proposed method had the highest accuracy for all classes. In particular, the detection accuracy for the infiltration and normal classes was significantly higher than that of the other methods.

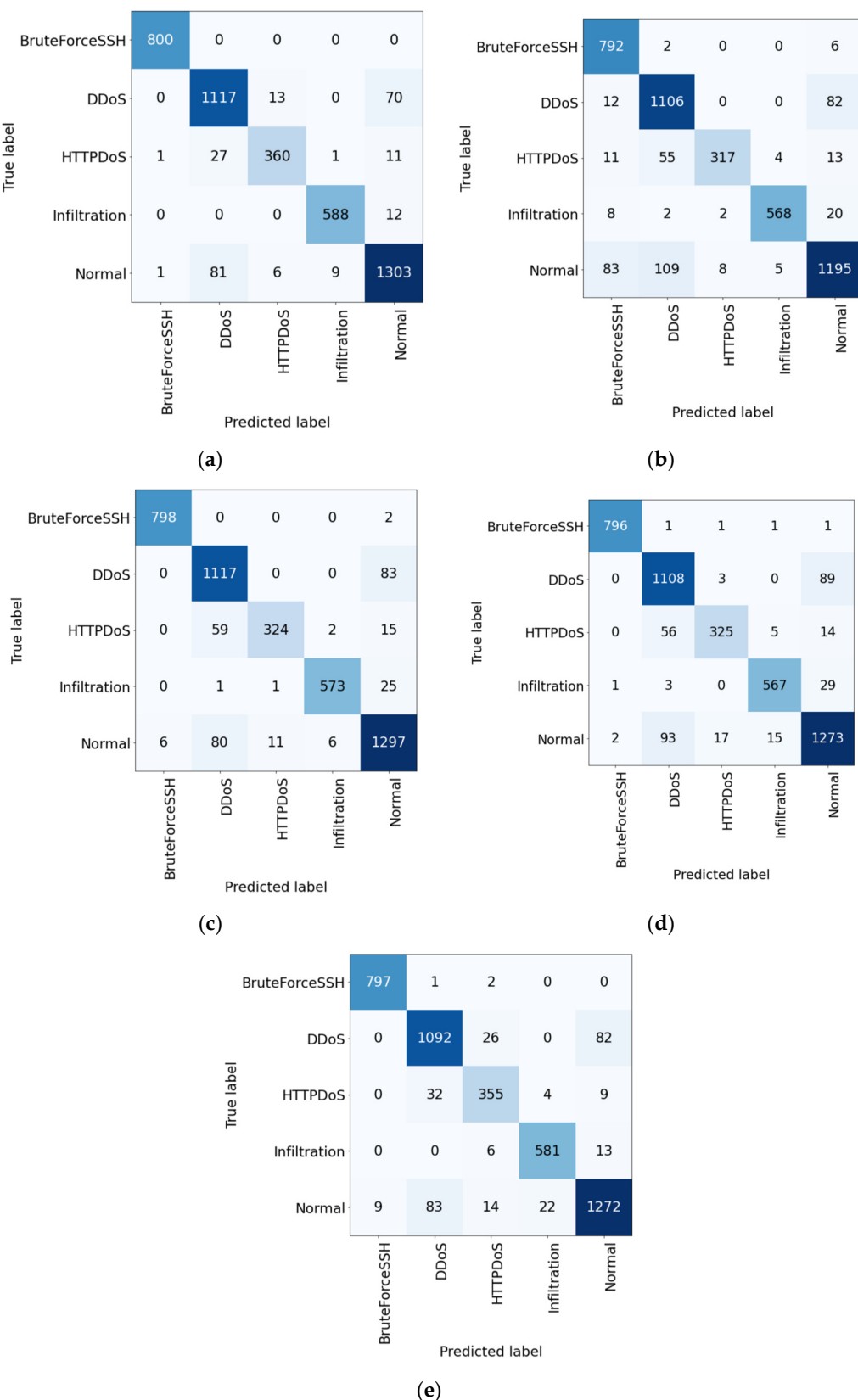

**Figure 14.** Confusion matrix for each algorithm using the ISCXIDS2012 dataset: (**a**) Proposed; (**b**) ELM; (**c**) DNN; (**d**) CNN; (**e**) TSE.

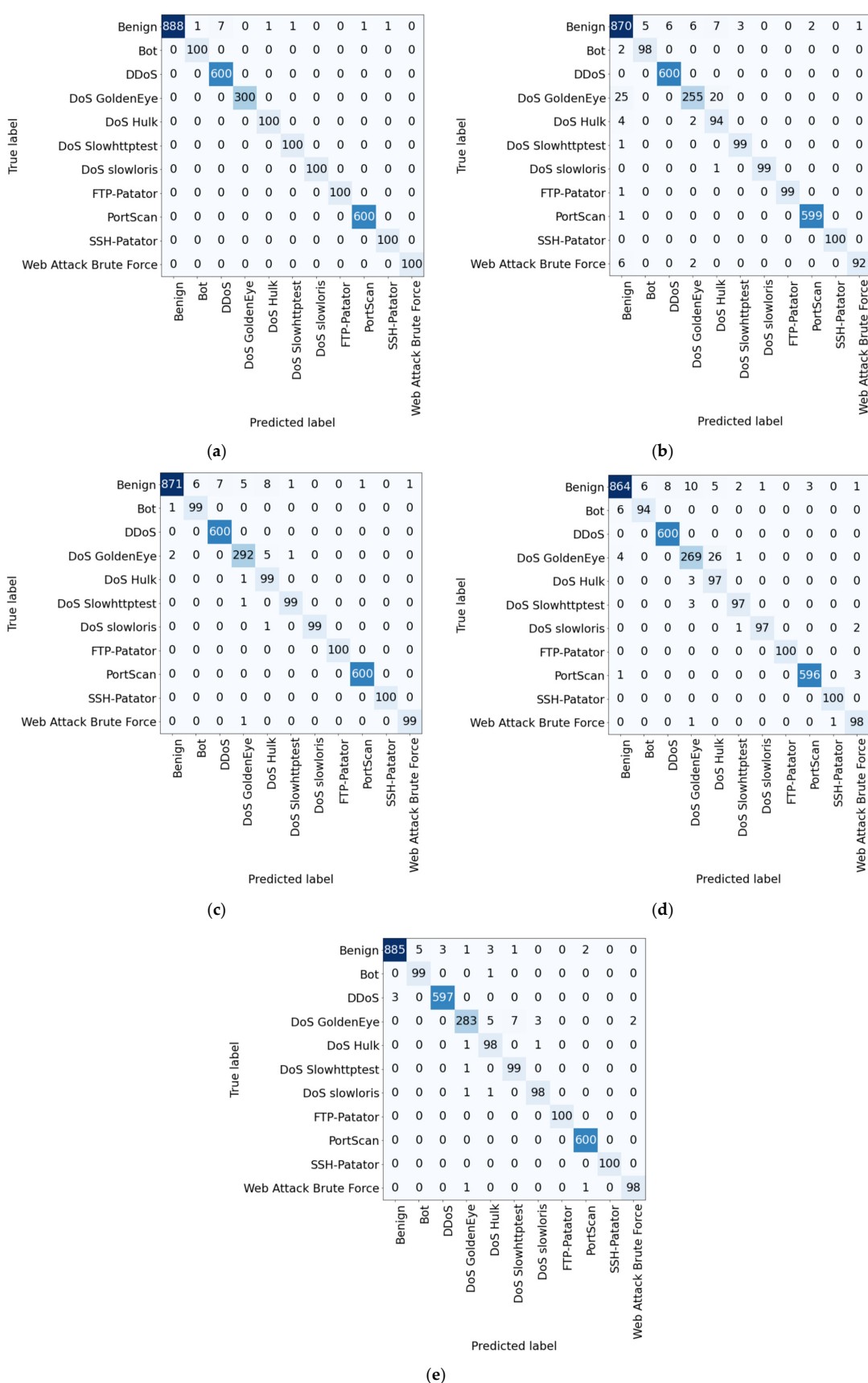

**Figure 15.** Confusion matrix for each algorithm using the CIC-IDS2017 dataset: (**a**) Proposed; (**b**) ELM;
(**c**) DNN; (**d**) CNN; (**e**) TSE.

It is important to accurately detect network intrusion in NIDS, but it is more important to accurately detect normal traffic as a normal class in order to provide stable network service to users. In this respect, the proposed NIDS shows the highest normal class detection accuracy compared to all existing NIDSs. Since the normal class has the largest number of sessions, it has the most diverse characteristics. Therefore, it is difficult to achieve high detection performance, but the proposed NIDS has high classification accuracy even for normal class because it uses all packet traffic to create features and detects network intrusion based on them.

As shown in Figure 15, the proposed method achieved the highest accuracy for all classes even for the CIC-IDS2017 dataset. Moreover, the performance for the benign and DoS GoldenEye classes was very high compared to the other algorithms. Therefore, Figures 13 and 14 confirm that the proposed method improves the overall detection performance as well as the detection performance for individual classes compared to the existing methods.

### 4.5. ROC Curve

Figures 16 and 17 show the receiver operating characteristic (ROC) curves for each dataset [28]. Regardless of the dataset, the performance on each class was superior to the existing algorithms. This confirms the superiority of the proposed method in terms of the overall performance and performance of each class compared to existing algorithms in various environments.

In particular, it can be seen that the detection accuracy for normal traffic is the highest in the ROC curves. The highest detection accuracy for normal traffic means that the proposed NIDS has the lowest negative impact on normal traffic for users. In addition, Figure 16 shows that HTTPDoS detection accuracy is very high compared to other NIDSs. Considering that DoS attacks are difficult to distinguish from normal traffic, the high accuracy of HTTPDoS detection means that the proposed NIDS accurately classifies even similar classes. It can be reconfirmed that the sophisticated structure of the deep learning model and the approach to generate features using all packet data are very effective in enhancing the detection performance of NIDS.

### 4.6. Classification Time

To compare the classification speed of each NIDS, the time required to classify the entire test dataset was measured for the two datasets. As shown in Figure 18, the experimental results show that the proposed NIDS has a higher total classification time than other NIDSs. Since the proposed DNN uses a deep learning model composed of dual LSTM, single LSTM, and DNN, the model complexity is very high compared to other NIDS classifiers.

Therefore, it can be assumed that the high classification time of the proposed NIDS is due to the complexity of these models. Slow classification speed is less important than classification accuracy because classification speed can be improved through parallel processing or feature engineering [29]. However, the low classification speed of the proposed NIDS needs to be improved because the NIDS that performs classification faster with the same hardware is advantageous in processing more traffic.

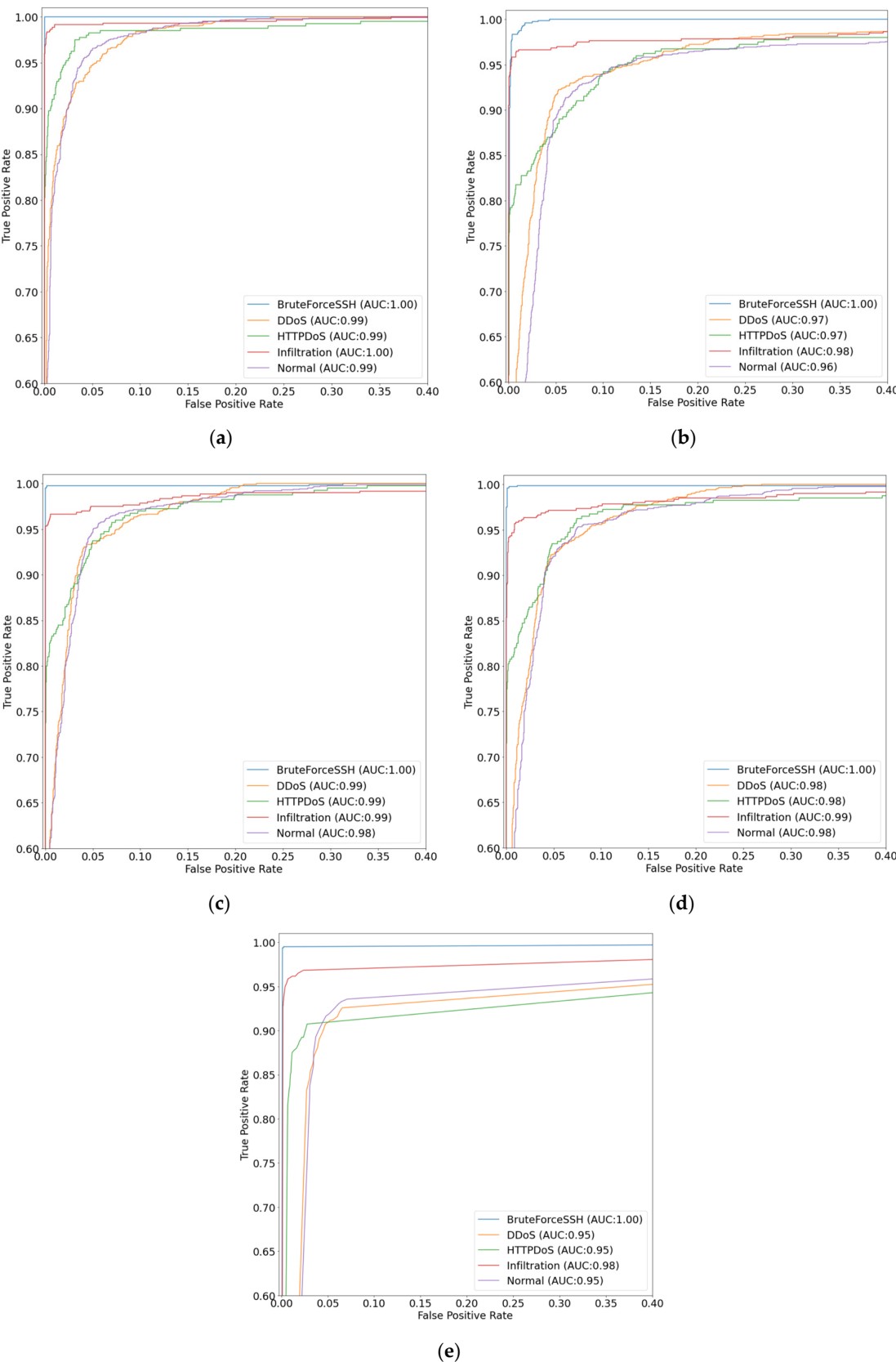

**Figure 16.** ROC curve of each class for each algorithm for the ISCXIDS2012 dataset: (**a**) Proposed; (**b**) ELM; (**c**) DNN; (**d**) CNN; (**e**) TSE.

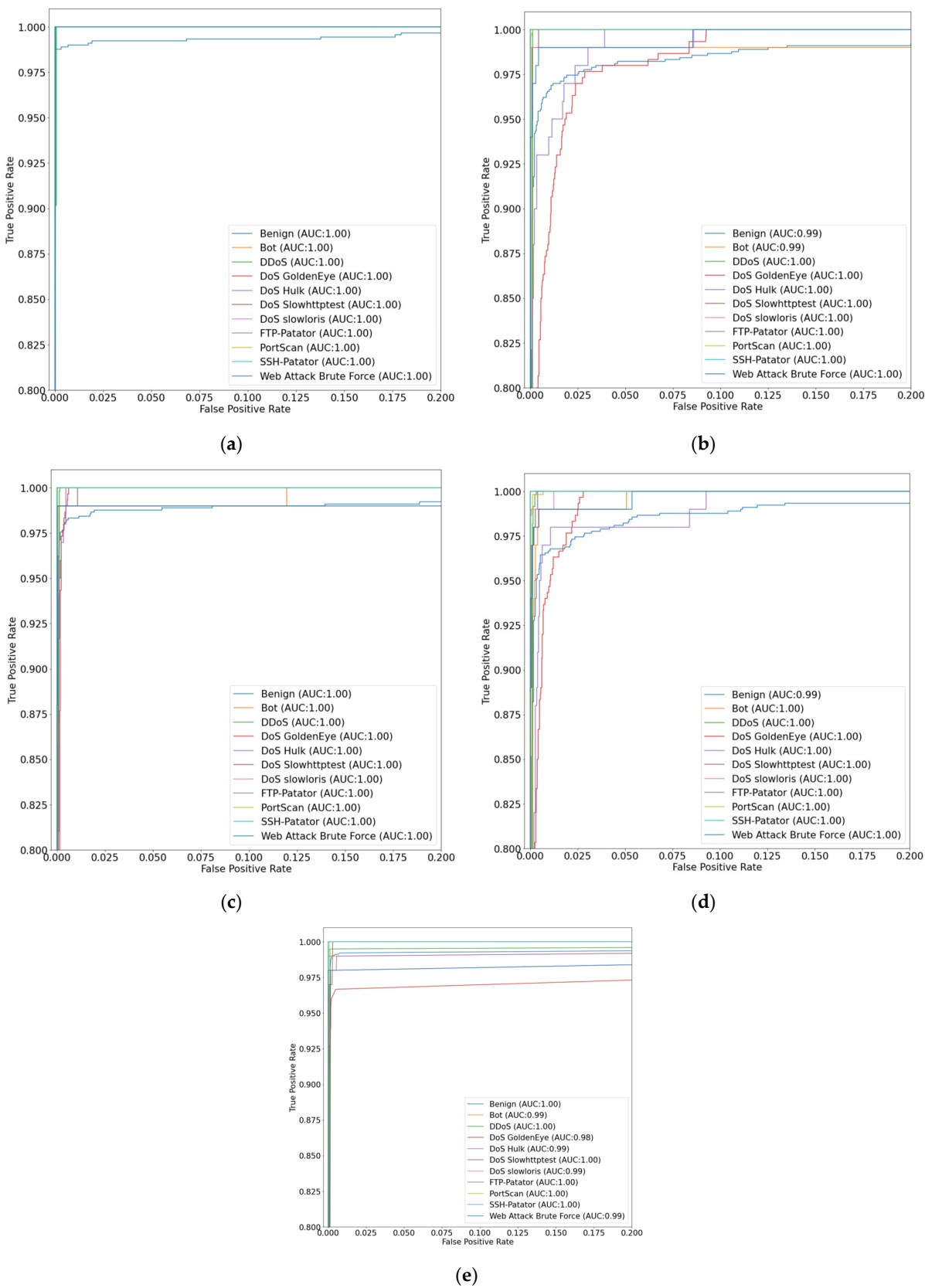

**Figure 17.** ROC curve of each class for each algorithm for the CIC-IDS2017 dataset: (**a**) Proposed; (**b**) ELM; (**c**) DNN; (**d**) CNN; (**e**) TSE.

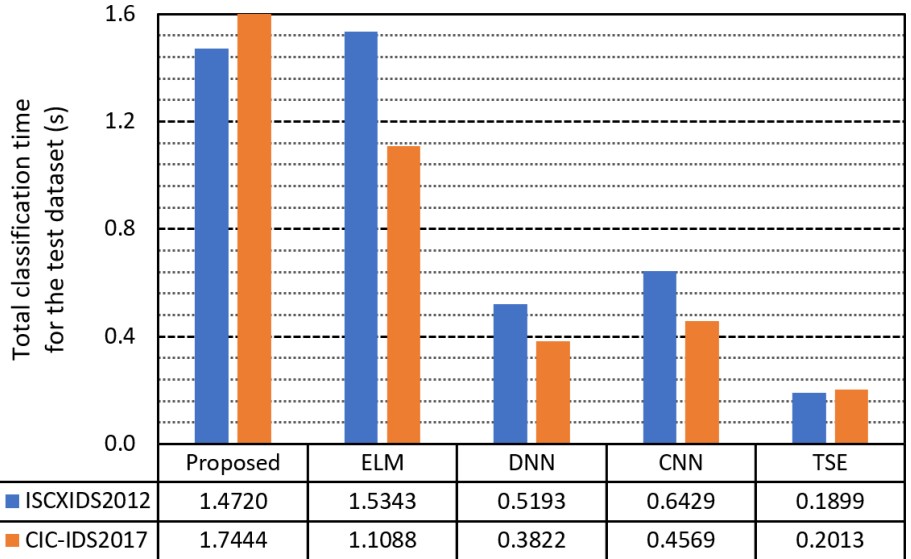

**Figure 18.** Total classification time of each NIDS for each test dataset.

## 5. Conclusions

The proposed NIDS algorithm is a hybrid method comprising a packet classifier composed of dual LSTM and a session classifier with a single LSTM. The existing machine learning-based NIDS detects intrusion using only some packets and some packets of a session owing to structural limitations, resulting in fatal flaws such as worsening detection performance and a high possibility of bypassing detection. On the other hand, the method proposed to address this problem was designed to detect intrusion by using all the data of a packet and as many packets as possible in a session.

Consequently, very high detection performance was achieved. In the performance evaluation, the proposed NIDS achieved the highest detection performance of 95.16% and 99.70% when the existing NIDS showed detection rates of up to 93.49% and 98.31% based on F1-score. In addition, the memory-optimized structure significantly improved the amount of memory required per session compared to existing methods, thereby minimizing total memory usage. Consequently, the number of concurrent sessions processed by NIDS can be significantly increased.

The proposed NIDS consists of a dual LSTM-based packet classifier, an LSTM-based session classifier, and a DNN that performs final classification. This complex model can greatly improve the detection accuracy and provides robust classification performance, but the classification speed decreases compared to other NIDSs. The slow classification speed can be partially improved by using parallel processing, but considering the large increase in traffic volume to be processed by modern NIDSs, high classification speed is very important from the practical point of view of NIDS. To solve this issue, various future work such as feature engineering and optimization of the deep learning model are required.

Through future work, the proposed NIDS can be extended and improved to be sufficiently used to protect enterprise-class networks. In particular, the proposed NIDS performs classification for each packet, so it is also possible to support real-time intrusion detection and will eventually be of great help in developing deep learning-based network intrusion prevention system.

**Author Contributions:** J.H. and W.P. wrote the paper and conducted the research. All authors have read and agreed to the published version of the manuscript.

**Funding:** This work was supported by the National Research Foundation of Korea (NRF) NRF2022R1A2C1011774.

**Institutional Review Board Statement:** Not applicable.

**Informed Consent Statement:** Not applicable.

**Data Availability Statement:** The datasets utilized in this paper are ISCXIDS2012 dataset (https://www.unb.ca/cic/datasets/ids.html (accessed on 22 February 2023) and CIC-IDS2017 dataset (https://www.unb.ca/cic/datasets/ids-2017.html (accessed on 22 February 2023).

**Conflicts of Interest:** The authors declare no conflict of interest.

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
