# Peer review of "Hierarchical LSTM-Based Network Intrusion Detection System Using Hybrid Classification"

_applsci, doi:10.3390/app13053089_

Round 1
Reviewer 1 Report
Comments to the Author (Please refer to the instructions above) The respected authors applied “The proposed NIDS algorithm is a hybrid method comprising a packet classifier composed of dual LSTM and a session classifier with a single LSTM.” Overall, the study is well-conducted and well-written. However, there are several serious concerns to be addressed before possible publication:
1. Please discuss related NIDS papers in the introduction and existing work section. and also add related references.
2. I think it is necessary to state the novelty of the study in the introduction section explicitly.
2. In figures 1(a) and 1(b), captions are repeated.
3. The first paragraph in section 3.1 (Motivation) is unclear, and please check for typos.
4. I suggest adding a flowchart to the before “Proposed algorithm” section so that future readers of the study get a better perspective of the work.
5. Conclusion is weak. It needs to be improved. And also add future work in the conclusion section.
6. What is the task behind this work? Explain it.
Reviewer 2 Report
The issues are:
-Abstract is not able to convey what is the technical contribution of this paper. I suggest to re-write it.
-Improve the quality of figures and explain those properly.
-The result section is weak and I suggest authors to add more results and compare those with the existing approaches.
-Although this paper is well written, there are still some typos in the current version
-The authors are expected to report the running time of the proposed algorithm in the revision.
-I suggest the author to be more precise in the description of the methodology and show how the methodology would achieve the stated objectives.
In the same field of interest, there are following good papers that will increase the technical strength of the article: Advances in security and privacy of multimedia big data in mobile and cloud computing, Phishing Dynamic Evolving Neural Fuzzy Framework for Online Detection “Zero-day” Phishing Email, PCNNCEC: Efficient and Privacy-Preserving Convolutional Neural Network Inference Based on Cloud-Edge-Client Collaboration
-I recommend the authors to thoroughly proofread the manuscript to correct all the typos.
Reviewer 3 Report
In this paper, the authors proposed a new deep learning-based network intrusion detection system that effectively uses the entire packet information using the hierarchical long short-term memory and achieves higher detection accuracy than existing methods.
The paper’s scope is within the scope of the journal, and it presents an original contribution. The abstract is somehow sufficient to give useful information about the paper’s topic. The proposed algorithm is somehow described and thoroughly illustrated. The paper is somehow well-structured and written, and the text is clear and easy to read. However, there are some comments we recommend the authors to do:
At the end of the abstract, it is worthwhile to present your best results as percentages in comparison to other existing methods.
In the introduction section, existing work section, or where appropriate, you need to write about intrusion detection in general and accordingly, you may need to cite and add the following two recent references:
Nwakanma, C.I.; Ahakonye, L.A.C.; Njoku, J.N.; Odirichukwu, J.C.; Okolie, S.A.; Uzondu, C.; Ndubuisi Nweke, C.C.; Kim, D.-S. Explainable Artificial Intelligence (XAI) for Intrusion Detection and Mitigation in Intelligent Connected Vehicles: A Review. Appl. Sci. 2023, 13, 1252. https://doi.org/10.3390/app13031252
Mishra, S. An Optimized Gradient Boost Decision Tree Using Enhanced African Buffalo Optimization Method for Cyber Security Intrusion Detection. Appl. Sci. 2022, 12, 12591. https://doi.org/10.3390/app122412591
Also, in the introduction or where appropriate, you need to cite and add the following recent reference regarding training very deep neural networks and Long Short-Term Memory (LSTM):
Abuqaddom, I.; Mahafzah, B.; Faris, H. Oriented Stochastic Loss Descent Algorithm to Train Very Deep Multi-Layer Neural Networks Without Vanishing Gradients. Knowledge-Based Systems 2021, 230, 107391. https://doi.org/10.1016/j.knosys.2021.107391
Moreover, in Section 3 (Proposed Algorithm), you need to mention whether your proposed approach suffers from vanishing gradients or not and explain why, where you can cite the above-mentioned reference (Oriented Stochastic Loss Descent Algorithm to Train Very Deep Multi-Layer Neural Networks Without Vanishing Gradients) regarding this issue.
In Section 3 (Proposed Algorithm), it is worthwhile to present your proposed algorithm as pseudocode as a table with a brief explanation.
In Section 3 and before Subsection 3.1 (Motivation), write one small overview paragraph about Section 3 and its Subsections 3.1 and 3.2.
The results figures (Figure 10–15) need more explanation and justifications. For example, it is not enough to say that our proposed algorithm is better than other existing algorithms, you need to explain why from regarding the algorithm's design point of view.
At the end of the conclusions section (Section 5), present your best results in terms of various performance metrics as values or percentages.
Round 2
Reviewer 2 Report
The core of the manuscript, Hierarchical LSTM-based Network Intrusion Detection System 2 using Hybrid Classification is an interesting topic for a broad audience, as it would trigger research from different areas. The paper concept is impressive, and also the novelty of this research work is significant. My feedback on this paper is given below:
· The abstract needs to be rewritten to point out significance and impact of the paper.
· More illustrations can be included in the revised manuscript, mainly to show more results and betterment of the presented approach.
· Add more discussion on the topic LSTM and Intrusion detection that includes quality research paper in the revised version. Author can refer following papers to improve the quality of article:
10.3390/electronics10030285
10.3390/s21196412
10.1109/IMCET53404.2021.9665641
10.1109/TransAI49837.2020.00017
10.3390/electronics11213529
· In Figure 1, the authors need to explain each and every component clearly.
· Avoid presenting with lengthy paragraph.
· Some more clarification regarding the motivation and challenges of the proposed approach, and how the prescribed scheme would overcome them must be added in the revised version.
· More recent references should be used or cited for better projection of the research problem and for absolute suitability of the journal's novelty.
· Future Scope and applications of the proposed scheme requires to be emphasized in the revised version.
· Some images are very small with poor quality; because of space, it would be good to zoom to the area the authors want to highlight.
· Please elaborate in detail about the proposed approach, focusing more on the relations between its components, as they are the core of the solution and need more justification for using them.
· The author needs to enrich the description of the system model by adding further details.
· Notations and acronyms used in this paper should be summarized in a table to organize this paper in a better way.
· The Author must talk about the computational overhead in the cost and complexity of the proposed work.
· I suggest adding more figures based on the proposed model/flowchart of the proposed model etc. to make strong to your paper.
· Results Section discusses the achieved experimental results. Such a section should be improved by reporting further details, especially for the discussion of Figures 9 and 10.
· Finally, a final proof-reading is highly suggested, in order to correct some typos.
